# The Cat Mandible (II): Manipulation of the Jaw, with a New Prosthesis Proposal, to Avoid Iatrogenic Complications

**DOI:** 10.3390/ani11030683

**Published:** 2021-03-04

**Authors:** Matilde Lombardero, Mario López-Lombardero, Diana Alonso-Peñarando, María del Mar Yllera

**Affiliations:** 1Unit of Veterinary Anatomy and Embryology, Department of Anatomy, Animal Production and Clinical Veterinary Sciences, Faculty of Veterinary Sciences, Campus of Lugo—University of Santiago de Compostela, 27002 Lugo, Spain; mar.yllera@usc.es; 2Engineering Polytechnic School of Gijón, University of Oviedo, 33203 Gijón, Spain; uo254292@uniovi.es; 3Department of Animal Pathology, Faculty of Veterinary Sciences, Campus of Lugo—University of Santiago de Compostela, 27002 Lugo, Spain; diana.alonso.penarando@rai.usc.es; 4Veterinary Clinic Villaluenga, calle Centro n° 2, Villaluenga de la Sagra, 45520 Toledo, Spain

**Keywords:** anatomy, feline, lower jaw, mandibular fracture, neurovascular supply, temporomandibular joint, tooth

## Abstract

**Simple Summary:**

The small size of the feline mandible makes its manipulation difficult when fixing dislocations of the temporomandibular joint or mandibular fractures. In both cases, non-invasive techniques should be considered first. When not possible, fracture repair with internal fixation using bone plates would be the best option. Simple jaw fractures should be repaired first, and caudal to rostral. In addition, a ventral approach makes the bone fragments exposure and its manipulation easier. However, the cat mandible has little space to safely place the bone plate screws without damaging the tooth roots and/or the mandibular blood and nervous supply. As a consequence, we propose a conceptual model of a mandibular prosthesis that would provide biomechanical stabilization, avoiding any unintended (iatrogenic) damage to those structures. The improvement of imaging techniques and a patient-specific prosthesis made of full biocompatible material are part of the future trends to improve patients’ recovery.

**Abstract:**

The cat mandible is relatively small, and its manipulation implies the use of fixing methods and different repair techniques according to its small size to keep its biomechanical functionality intact. Attempts to fix dislocations of the temporomandibular joint should be primarily performed by non-invasive techniques (repositioning the bones and immobilisation), although when this is not possible, a surgical method should be used. Regarding mandibular fractures, these are usually concurrent with other traumatic injuries that, if serious, should be treated first. A non-invasive approach should also first be considered to fix mandibular fractures. When this is impractical, internal rigid fixation methods, such as osteosynthesis plates, should be used. However, it should be taken into account that in the cat mandible, dental roots and the mandibular canal structures occupy most of the volume of the mandibular body, a fact that makes it challenging to apply a plate with fixed screw positions without invading dental roots or neurovascular structures. Therefore, we propose a new prosthesis design that will provide acceptable rigid biomechanical stabilisation, but avoid dental root and neurovascular damage, when fixing simple mandibular body fractures. Future trends will include the use of better diagnostic imaging techniques, a patient-specific prosthesis design and the use of more biocompatible materials to minimise the patient’s recovery period and suffering.

## 1. Introduction

As seen in the first part (The cat mandible (I) [1]), a deep knowledge of the functional anatomy of the feline mandible will be the basis for interpreting the diagnostic images that, in combination with symptoms, will help to achieve an accurate diagnosis about the current pathology. Afterwards, the clinician should come to a decision and a treatment has to be prescribed. A meticulous therapeutic decision-making process is essential to choose among alternatives and analyse consequences, as some treatment procedures include surgery.

Hence, the present review will be focused on how to solve different pathological situations of the feline mandible and temporomandibular joint, such as luxation or fractures. In addition, we will insist on the importance of keeping the integrity of the tooth roots and the mandibular canal neurovascular bundle, as they could be damaged during fracture repair.

## 2. Temporomandibular Joint Luxation

The temporomandibular joint (TMJ) is characterised by the presence of a whole intra-articular disc, although is very thin in the cat and in the dog. Many elements are related to this synovial condylar joint and gaining a comprehensive knowledge of this joint is required to correctly interpret its diagnostic images. This ability is essential in order to make a diagnosis and to achieve good results in the management of different conditions, minimising the incidence of surgical iatrogenic lesions. While TMJ caudal luxation might occur with fractures in the retroarticular process, mandibular fossa or mandibular head, rostral mandible movement is more common [2]. Unilateral rostrodorsal luxation of the mandibular condyle causes the lower jaw to shift laterorostrally to the luxation-opposite side and the inability of the cat to close its mouth fully due to tooth-to-tooth contact. This distinguishes it from open-mouth jaw locking, where the mouth is held wide open with no contact between mandibular and maxillary teeth [3]. The unilateral rostrodorsal luxation of the mandibular condyle could easily be reduced by placing a pencil between the maxillary fourth premolar (PM4) and the mandibular first molar (M1) on the affected side only and closing the mandible against the pencil while simultaneously relieving the jaw caudally (easing the condyle from the articular eminence). However, this treatment is contraindicated in animals with open-mouth jaw locking [3].

Open-mouth jaw locking is characterised by an inability to close the mouth that usually results from fixed mandibular coronoid process displacement lateral to the ipsilateral zygomatic arch and abnormal contact pressure between these two structures [2]. The history is important in the diagnosis, as it is usually observed after animals have yawned, groomed or vocalised [2]. Physical findings of a wide-open mouth and palpable coronoid process superficial to the zygomatic arch help to distinguish open-mouth jaw locking from TMJ dislocation [2]. Manual reduction is the first-line treatment method in open-mouth jaw locking, secondary to coronoid process zygomatic arch interlocking and temporomandibular dislocation. Resolution may be spontaneous or require manual correction, but recurrence is possible [2]. However, and according to Reiter and Lewis [3], the treatment consists of two phases: an acute treatment (under sedation) consisting of opening the jaw even further to release the coronoid process from the lateral aspect of the zygomatic arch, and then closing the mouth. A tape muzzle should be placed until definitive surgery is performed. This consists of a partial resection of the coronoid process, partial resection of the zygomatic arch, or a combination of both [3].

## 3. Mandibular Fractures

### 3.1. General Considerations

Mandibular fractures are commonly seen in practice, comprising up to 6% of all fractures in dogs, and 11–23% of all fractures in cats, according to Glyde and Lidbetter [4]. They occur more frequently than maxillary fractures [3]. The majority of mandibular fractures in cats may result from road traffic accidents, fighting injuries or falls from heights [4,5,6] and, unfortunately, also due to human abuse [7]. A traumatic aetiology commonly involves serious concurrent injuries requiring prompt clinical attention, mainly to the brain, maxilla and chest [8]. Management of life-threatening injuries and normalisation of patient physiology is required before surgical stabilisation of mandibular fractures [8]. Less commonly, pathological mandibular fractures may occur secondary to periodontal, oral neoplasia or metabolic disease, and iatrogenic fractures can also occur during dental treatment [3,4]. Although various types of injuries and trauma are typically responsible for fractures of the upper (maxilla) and lower jaw, certain risk factors may predispose a cat to fractures, including oral infections (e.g., periodontal disease, osteomyelitis), certain metabolic diseases (e.g., hypoparathyroidism) and congenital or hereditary factors resulting in a weakened or deformed jaw [9].

Independently of the aetiology, in cats, and according to Umphlet and Johnson [10], mandibular fractures accounted for 14.5% of all fractures seen in a total of more than 500 cat specimens (*n* = 517). Symphyseal fractures were the most common (73.3%), followed by fractures of the body (16%), condyle (6.7%) and coronoid process (4%). Complications developed more commonly in cats with multiple or open fractures. Clinical union occurred after an average of 6 weeks (range 3–12 weeks) for symphyseal fractures, 10 weeks (range 8–16 weeks) for body fractures, 6 weeks for coronoid fractures and 6 weeks (range 4–8 weeks) for condylar fractures [10]. In contrast, in dogs, fractures in the premolar region are significantly more frequent than in other regions [11]. Umphlet and Johnson [11] also reported that fractures in the rostral portion of the mandible had shorter average time to clinical union than did other mandibular fractures. However, the average time to clinical union for fractures in the caudal portions of the mandible was longer than that currently reported [11]. Nonetheless, overall prognosis depends on type, extent, location of trauma, quality of home care and selection of treatment modality [9].

Accordingly, Little [12] reported that mandibular fractures in cats are typically located in the area of mandibular symphysis or the mandibular ramus (fractures of the condylar process or coronoid process). The midportion of the mandibular body is less frequently fractured in cats.

A non-invasive approach should receive primary consideration, and an invasive option is employed only if non-invasive treatment is insufficient or impractical [13]. Tape muzzling is a non-invasive and inexpensive treatment option for mid-body and caudal mandibular fractures and for TMJ luxation and open-mouth jaw locking after manual correction [13]. Tape muzzles could be used as temporary, definitive or adjunctive therapy. This is a good method in cases of pathological fractures or where the bone is very porous and will not support a fixative device. Where fractures are stable, this is also a good technique [14]. It may be curative for mandibular fractures in immature, adolescent and young adult animals with good bone healing capacities. In addition, muzzling allows some TMJ movement, thus reducing the risk of ankylosis between fractured bones in that area [13].

In the cat, when the jaw is immobilised to allow healing of the fracture, the mouth should be kept open no less than 5 mm and no more than 10 mm (as measured between the incisal edges of the maxillary and mandibular incisors) to allow for the tongue to protrude and lap water and a slurry diet. If the mouth is open too far, it will result in difficulty in swallowing [13]. This immobilisation of the mandible, to limit oral opening, could be done with a tape muzzle, fixation with composite spanning the ipsilateral canine teeth, or through labial buttons placed with suture material [15].

Southerden et al. [16] reported that there is a low level of asymmetry between contralateral mandibles in cats, allowing the use of a mirror image of an intact mandible for planning and evaluating the accuracy of fixation of a contralateral mandible. The most consistent measurement among 27 specimens was the lateral ramus inclination angle. However, the least consistent measurements were ramus height and jaw width at the mental foramen [16]. This type of study may facilitate the development of standardised pre-contoured locking plates for cat mandible repair.

Regarding mandibular biomechanics, as reported by Spodnick and Boudrieau [17], a continuity of tensile to compressive stresses exists from one side of the bone to the other during bending stress. Maximal tensile stresses are present at the oral (alveolar) surface and maximal compressive stresses at the aboral (ventral) surface; therefore, distraction is created at the oral margin. These bending moments increase from caudal to rostral; furthermore, shear forces are maximal at the ramus, while rotational forces are most prominent rostral to the canine teeth and maximal at the mandibular symphysis [17]. Taking these biomechanics into account would be of great help when fixing a mandibular fracture. Consequently, invasive jaw fracture repair techniques (osseous wiring, external skeletal fixation and bone plating) should be carefully planned and be accompanied by dental radiography, both intra- and post-operatively [16].

### 3.2. Symphyseal Fractures

Mandibular symphysis fractures are the simplest and are best treated with cerclage wire [4]. Palpable instability of the symphysis is not pathognomonic for traumatic symphyseal separation, as instability may result from periodontal disease, laxity of the ligamentous attachment, neoplasia, or fracture of the mandible [18]. Relative to the placement of a circumferential wire for mandibular symphyseal fracture repair, and according to Glyde and Lidbetter [4], it is necessary to place a hypodermic needle to act as support, making a hole from the oral cavity (from the caudolateral edge of the canine tooth) and leave the wire end at the level of a skin incision in the intermandibular space; however, on the other side, the needle is introduced from the bottom to the oral cavity to allow removing the needle once the wire is in place. This is a logical sequence, taking into account that the wire should make a loop to press the two mandibles together and reassure the mandibular symphysis. Make sure that the incisor teeth remain in alignment; otherwise, step defects can be generated [15]. The wire should be removed once union has been achieved [4].

Parasymphyseal fracture is a common iatrogenic injury during extraction of the mandibular canine tooth in cats [15]. The fracture may occur due to pre-existing periodontal or endodontic disease, insufficient preparation prior to extraction, or excessive force used by the operator, or a combination [15,19]. It turns into an important pathology that remains undetected if postoperative radiographs are not obtained, as the fracture often is non-displaced [19]. As a recommendation of good practice, Hoffman [20], a diplomate of the American Veterinary Dental College and board-certified in veterinary dentistry, advised that two dental X-rays should be always taken: (1) before extractions (this will allow the veterinary dentist to assess the health of the bone and the anatomy of each tooth, including its roots, taking into account that advanced dental disease contributes to bone loss and increased risk of iatrogenic trauma), and (2) after an extraction to ensure that the entire dental root has indeed been removed. In addition, the client should be informed upfront that a jaw fracture is a possibility, to avoid difficulties regarding iatrogenic jaw fractures secondary to tooth extraction. Furthermore, her advice is that once you are faced with an iatrogenic mandibular fracture, the case should be always referred to a specialist [20]. Parasymphyseal fractures could be treated with circumferential wire [15].

### 3.3. Body Fractures

When assessing the mandibular body for fractures, the direction of the fracture and the location of the fracture in relation to the dental roots should be evaluated. From a biomechanical point of view, fractures could be simple (in which the fracture line is perpendicular to the long axis of the mandible), or oblique fractures, that may be described as favourable or unfavourable according to the difficulty of immobilisation. This distinction results from the forces that the muscles of mastication place on the mandible as they either compress (favourable) or distract (unfavourable) the fracture segments [15,21] (Figure 1). Hence, a fracture that travels dorsocaudal to ventrorostral is considered favourable, whereas a fracture that travels dorsorostral to ventrocaudal and distracts the fracture fragments is considered unfavourable [15,21,22]. Favourable fractures compress because the upward pulling of the masseter and temporalis muscles and the downward and caudal pulling of the digastricus will hold the fracture segments in apposition, to a large extent. They are relatively stable, and stabilisation of the tension surface may be all that is required for bone healing [21]. In contrast, in unfavourable fractures, the alveolar crestal bone is considered the tension surface, while the ventral cortex is considered the compression surface [21], and the muscular forces will lead to considerable displacement of the fracture segments [3].

As far as fracture repair is concerned, it is extremely important to take notice of a large percentage of the mandibular body being occupied by deep dental roots—between 50% and 70% of its dorsomedial depth [4]—and the neurovascular structures (along the mandibular canal), facts that largely limit the areas for safe implant/screw placement (i.e., for plate fixation). This is particularly true in cats, where only a small amount of ‘free’ bone exists rostrally to the first premolar and caudally to the molar tooth [4]. Radiologically, the mandibular canal can be placed parallel and just coronal to the ventral cortex (the radiopaque bone in the ventral mandibular body margin) as a thick, horizontal radiolucent line in close apposition to the mandibular premolar and molar dental root apexes [18,23]. In that sense, Bellows [21] advises “Unless you’ve had advanced training, avoid plating jaw fractures for fear of compromising dental roots. Also avoid placing intramedullary pins into the mandibular canal. The mandibular canal carries the neurovascular structures—it is not an intramedullary canal”. Accordingly, one myth stated by Hoffman et al. [24], to dispel common misconceptions relative to “intramedullary pins in the mandibular canal are an option for treatment of fractures of the mandibular body”, said that experimentally, inserting pins into the body of the mandible results in delayed healing and considerable damage to many dental roots, whereas clinically, malalignment is a common complication.

As stated by Lantz [25], the principles of facial repair include (1) restoration of occlusion and anatomic reduction, (2) stable fixation to neutralise detrimental forces on the fracture line(s), (3) preservation of blood supply, (4) preservation of soft tissue attachment to bone fragments by gentle tissue manipulations and minimal tissue elevation, (5) avoidance of iatrogenic dental trauma, not injuring the dental roots, and (6) extraction of diseased teeth at the line(s). The selected method of repair should provide occlusion and, ideally, rigid stability of all major fragments. In addition, the device should allow immediate return to oral function, be lightweight, economical and not cumbersome for the patient. However, Spodnick and Boudrieau [17] reported that removal of teeth may increase complications due to disruption of the blood supply and iatrogenic trauma to the adjacent tissues, including further displacement of the bone fragments, elimination of occlusal landmarks useful in realigning bone segments to allow functional occlusion, elimination of available structures for use in the fixation of bone fragments and creation of a large bony defect adding to the difficulty of reduction and stabilisation. Preservation of teeth involved within a line in a mandibular fracture has also been reported to have a favourable prognosis if optimal reduction and stabilisation of the jaw has been achieved. Therefore, removal of teeth is not advised unless the teeth involved are fractured (even here universal removal is not recommended if the tooth contributes to stabilisation and if the fracture of the tooth does not involve the root). Nonetheless, advanced periodontitis or periapical abscessation are situations in which in-line teeth should be removed since they have contributed to pathological fracture of the mandible [17].

Mandibular body fractures may be treated in various ways. Favourable body fractures could heal using conservative methods to get stabilization of the tension surface [15,21]. Internal fixation with interfragmentary wires is indicated for simple mandibular body fractures, such as unfavourable fractures. The primary function of osseous wiring is to reduce fractures and prevent their displacement by the passive function of the muscles of mastication. The first rule of osseous wiring is to do no harm or to attempt to avoid injury to other structures within the jaws, such as the mandibular canal, roots of teeth and the periodontal ligament. The second rule is always to attempt to re-establish normal functional occlusion [15], as the mouth must be able to shut after surgery. To repair an unfavourable fracture, two intra-osseus wires are needed (one dorsally and the other ventrally) between the two fragments [15], or a triangular method could be used instead, consisting in one hole rostrally to fix two wires in a perpendicular arrangement coming from two holes placed caudally to the fracture. The dorsal wire should have a horizontal disposition and the second wire provides additional stability and prevents shear or rotation of the fragments around the primary wire [8]. Additionally, interdental wiring could also be used to increase the fracture stability [15].

Another option to treat unfavourable fractures is by using internal fixation with conventional bone miniplates and screws. The advantages of internal rigid fixation in the treatment of mandibular fractures are the accurate restoration of normal anatomy and occlusion and rapid return to normal function [16]. Nevertheless, every effort should be made to avoid injury to the roots and periodontal ligament of the teeth and the mandibular canal during placement of the anchorage holes through the bone [15].

The mandible is not an easy bone in which to use plating techniques for stabilisation of fractures. According to Higgins [8], at least two plates are required in the mandible. One is placed along the alveolar surface to resist bending and act as a tension band. A secondary plate is required on the ventral surface to resist shear and rotation. However, the anatomy of the mandible in the cat precludes placement in a biomechanically advantageous position of the minimum two pins in each fragment without the risk of compromising either the dental roots or the mandibular canal [4,26]. It should be noted that dental damage may result in pain, infection, tooth death, periapical lesions and ultimately failure of fixation because of persistent infection and inflammation at the fixation site [4]. The application of other metal implants should be undertaken noting the following: large teeth occupy 70% of the depth of the bone, damage to vessels and nerves within the mandibular canal should be avoided and feeding tubes should be considered [27]. Thus, bone plates can be placed in the caudal toothless part of the mandible at the junction of the body and the ramus [4].

The article by Greiner et al. [28] biomechanically evaluates two plating configurations (using one or two internal fixation plates) to fix a simple transverse caudal mandibular fracture, in particular, shown in Figure 1 (and onwards), in which the legend states that the ideal region for miniplate application (where the bone is able to support internal fixation) in the cat mandible is depicted in blue. However, the blue colour coincides all along with the path of the mandibular neurovascular bundle, the main blood and sensory supply. They considered the nonideal region for miniplate application to be the area occupied by the dental roots, even though other figures show superimposition of the screw holes with the dental roots. Biomechanically, it could be an acceptable model, but it must be extremely painful for the cat patient. This happens because, usually, the mandible is considered like any other long bone, without taking into account its particularities: (1) the jaw has specialized structures, the teeth, whose roots are inserted in the alveoli and kept in position thanks to the periodontal ligaments, and (2) it does not have a medullary canal, but an inner canal that runs along the jaw (from caudal to rostral) providing blood supply and sensitive innervation, not only to the bone, but also the teeth and other related structures, such as mucosa and gingiva. Thus, if any of these structures become damaged, it does not matter that the prosthesis would be acceptable from a biomechanical point of view, because it is functionally unsatisfactory. Hence, it would be advisable that researchers, clinicians and surgeons be aware of the importance of maintaining the integrity of the neurovascular supply and dental roots in order to achieve a favourable outcome.

Plate placement on the ventral (compression) border of the mandibular body, which avoids the neurovascular structures of the mandibular canal, increases the load on the plate. Mono-cortical application of bone plates to the mid-buccal surface of the mandible has been recommended to reduce the risk of iatrogenic damage [4]. Consequently, standard bone plates in accordance with the feline mandibular size must be used. Nowadays, there are many companies producing osteosynthesis systems with titanium microplates (up to a minimum thickness of 0.6 mm to fix with 1 mm diameter screws). These sizes, and similar, are recommended for the repair of feline mandibular fractures. The plate should therefore be placed on the ventral surface to avoid these structures, although the tension side is the oral side. Plating should only be considered for simple fractures that can be very accurately reduced. Any malalignment may lead to dental malocclusion, which would be difficult to correct. Locking plates may be more useful than non-locking implants as, if plate contouring is not perfect, they should not distort the fragments. The most common complications of surgical repair are malocclusion and osteomyelitis [15,21].

In the case of comminuted or open fractures, in which soft tissue wounds prevent the use of internal fixation, an external fixation with pins and a mandibular bumper bar is appropriate for their reduction [26].

According to Spodnick and Boudrieau [17], the management of simple fractures (large fragments without comminution) could be done with routine induction and endotracheal intubation per os for anaesthetic maintenance and surgery. In these instances, anatomical re-alignment and reduction of the fragments, rather than dental occlusion, is used to determine the accuracy of surgical reduction. Alternatively, in cases of severely comminuted fractures or those with bone loss, dental occlusion must be used to access the accuracy of the surgical reduction. In these cases, the endotracheal tube impedes the assessment of occlusion by preventing full closure of the mouth and must be replaced to bypass the mouth (endotracheal intubation via pharyngotomy). Hence, occlusion is used to determine the accuracy of the reduction when comminution or gaps in the bone are present. Simpler fractures may be reconstructed anatomically. As usually one side of the head/face is more severely injured, a reasonable approach is to repair the side with the simpler fractures first. It is highly recommended to repair the mandible from caudal to rostral, with symphyseal separations secured as the final step [17]. Interestingly, the ventral approach to each mandible facilitates exposure and bone fragment manipulation, including the ability to perform an accurate reduction and stabilisation. Consequently, the patient should be placed in dorsal recumbency to get the head in the ventral approach, fixing it by taping the maxilla to the table [17].

### 3.4. Fractures of the Ramus

Fractures of the mandibular ramus are relatively stable because the surrounding muscles usually prevent gross displacement of the fracture segments [3]. Condylar process fractures often heal as pain-free and functional non-union without surgery, but comminuted fractures could result in a TMJ ankylosis in immature and young cats [3]. Ramus fractures in dogs and humans are frequently stabilised using internal rigid fixation with plates and screws, providing accurate reduction and good construct stability [16]. But in cats, this is complicated because of their small size, the need for greater contouring of implants, difficulty in positioning small fragile fragments of bone during the application of implants and the very small cross-sectional surface area of bone at the fracture sites, making accurate anatomical reduction challenging [16]. However, Southerden et al. [16] proposed the development of a small range of standard pre-contoured locking plates for the fixation of caudal mandibular fractures in cats due to the small variation in shape and size of mandibles between animals (among 38 specimens).

### 3.5. Impairment of the Nervous Supply

It should be taken into account that more than the upper two-thirds of the body of the mandible is occupied by dental roots. The ventral third includes the mandibular canal, containing the inferior alveolar nerve and associated blood vessels, and the inferior alveolar artery and vein. The inferior alveolar nerve provides sensory innervation for the teeth and leaves the bone through three mental foramina as the mental nerves. These nerves are sensory to the soft tissues of the rostral part of the mandible. The blood vessels in the mandibular canal supply all the teeth by the way of small dental branches (*rami dentales*) entering the apical foramina and the bone itself [17].

In humans, according to Misch and Resnik [29], and regarding nerve injuries after dental implant procedures, traumatic and iatrogenic nerve complications may involve total or partial nerve resection, crushing, stretching or entrapment injuries. As a consequence, the resulting sensory deficits may range from nonpainful minor loss of sensation to a permanent and severe debilitating pain dysfunction. Regarding oral and maxillofacial tumours in cats, Little [12] shows an example of mandibulectomy (removing left total and right partial mandibles), in which “the inferior alveolar artery and vein entering and exiting the mandibular canal through the mandibular foramen at the medial aspect of the mandible are ligated and transected”. But nothing is said about the alveolar mandibular nerve, as it is supposed that its resection might become very painful (even post-resection) as it has a sensory component. The point is that humans can express and describe their loss of sensation or pain, but what about cat patients after mandibular surgery? It is reasonable to assume that they can also suffer from the same type of impairment, but they cannot describe their sensations and could be suffering from severe debilitating pain dysfunctions or neuropathic pain and not even be willing to eat normally by themselves again.

In mandibular fractures, besides mandibular reconstruction, the integrity of the mandibular nerve is a fundamental aspect to take into consideration, since it has a sensitive, in addition to the motor, component. Consequently, it is not recommended to proceed to cut the mandibular nerve just to remove it, given that it can induce trigeminal ganglion degeneration, as reported by Gobel and Binck [30] following pulp removal in cats, inducing degenerative changes in primary trigeminal axons and in neurons in the *nucleus caudalis*.

Mandibular fracture types, their incidence and treatment methods are summarized in Table 1.

## 4. Prosthesis Proposal to Fix a Simple Fracture of the Mandibular Body

Taking into account all the concerns previously revealed, we propose a fixation method to repair simple fractures of the mandibular body that will provide acceptable rigid biomechanical stabilisation yet avoiding dental root and neurovascular damage.

The current prosthesis design proposal was made by using the design program Solidworks^®^. This computer-aided design (CAD) model was built starting from the three-dimensional (3D) model of a real jaw. A cloud of points, attained after laser scanning a real jaw, has been post-processed to obtain the 3D model of the cat mandible used. Meshlab and Cloud Compare software were used to post-process the data obtained by laser scanning.

The proposed prosthesis (Figure 2, Figure 3, Figure 4 and Figure 5) is a conceptual design that, if referred to the technological maturity level scale known as Technology Readiness Levels (TRL), would be equivalent to a TRL2, which corresponds to the formulation of a conceptual solution without yet testing experimentally. Hence, seven other TRL levels would lay ahead, including research in the laboratory environment (up to TRL4), then, the simulation environment (TRL5–6), and finally, the real environment (TRL7–9).

We think it could achieve good results in further TRL levels as it has three fixation points with small screws and the fourth is like a folded tab for fastening the ventral edge, therefore avoiding drilling the mandibular canal. The design with a two-sided ‘Y’ was chosen in a way that the screw positions do not imply any risk of perforating dental roots or damaging the neurovascular support. This shape also allows keeping safe the mental nerve that goes through the main mental foramen and also avoiding the caudal mental foramen.

The suitable material to implement this prosthesis, given its high requirements of resistance and biocompatibility, is titanium. The optimal manufacturing method would be ‘additive manufacturing’ (a transformative approach to industrial production that enables the creation of lighter, stronger parts and systems), i.e., metal 3D printing, since the prosthesis is a unique piece of small size, and up to now, this is the only method that makes production economically viable. In addition, given that its geometric complexity does not pose problems for its manufacture, this method is considered the best option.

In order to promote good adhesion of the prosthesis to the bone, and trying to avoid damage to the bone, the surface of the prosthesis should have a high roughness (undulations, geometric patterns with protrusions or some geometry with an equivalent effect). Thus, the effective area of contact with the bone (i.e., the actual contact area) would be minimised while increasing the friction force of the prosthesis with respect to the bone, which greatly reduces the possible relative slipping that may exist between the elements.

### 4.1. Calculations

Considering that at the time of application of the prosthesis, the jaw is immobilised, and that once the bone has healed the stresses of the prosthesis are greatly reduced, the function of the prosthesis at the beginning consists fundamentally in ensuring that the bone fragments are correctly and firmly positioned. A calculation of resistance based on a bite force of 10 kg is considered conservative.

### 4.2. Flexural Strength

Considering this force and that the central part of the prosthesis is the weakest part (its most critical section), which is placed 2 cm away from the canines, it is calculated that it supports a moment (*M*) of:M=F×d=50 N×25 mm=1250 N·mm
where:*F* is the biting force (N)*d* is the distance between the point of application of the force and the point of evaluation (the maximum distance to obtain the most critical effort) (mm)

Note that the force considered is 5 kg, since it is assumed that half of the effort is carried by the other jaw.

Taking into account the moment calculated previously, a first approximation of flexure resistance could be obtained dividing the moment by the bending moment resistance (Wb), as is shown in the next equation [31], with a considered thickness of 1.5 mm and width of about 3 mm, there is a maximum bending stress of:(1)$$\sigma^{max}=\frac{M}{\frac{b\cdot h^{2}}{6}}=\frac{1250\;N\cdot \mathrm{mm}}{\frac{1.5\;\mathrm{mm}\times {\left(3\;\mathrm{mm}\right)}^{2}}{6}}=555.56\;\mathrm{MPa}$$

Equation (1): Maximum traction tension considering only flexion efforts.

where:*b* corresponds to thickness value (mm)*h* corresponds to height value (mm)*M* corresponds to flexure moment value (N·mm)*σ* corresponds to tension value (MPa)

The value of the bending moment resistance is obtained considering a squared section of flexure; if the design of the prosthesis changes, as a result of an irregular section chosen, the finite element method (FEM) would be needed to calculate the maximum bending tension.

As it can be seen, the stress obtained is considerably lower than the elastic limit of titanium. Thus, it could be considered a valid measurement. However, a balance should be struck between material resistance and physiology. Regarding the field of material resistance, it is advisable to make the prosthesis with a width as large as possible to achieve greater rigidity. However, this is inadvisable from the physiological point of view. Therefore, it is a question of achieving a balance between good rigidity and comfort.

### 4.3. Shear Resistance of Screws

Screws with a diameter of 1 mm have been considered, and taking into account the previous data and the conservative hypothesis that all the force is supported by one of them, the following shear stress is obtained [31]:(2)$$\tau^{max}=\frac{4}{3}\times \frac{F}{\pi r^{2}}=85\;\mathrm{MPa}$$

Equation (2): Maximum shearing tension considering bite force

where:*τ* corresponds to shear tension in screws (N/mm)*F* corresponds to bite force (F)*r* corresponds to screw radius (mm)

The resulting stress is considerably lower than the shear strength of titanium, which means that the diameter of the screws can be even smaller.

Depending on the shape or position of the screws, a deeper calculation based in the FEM would be needed.

As previously indicated, the prosthesis shown is purely conceptual, and although an idea of the thickness and width that might be needed has been given, the calculations must be reviewed in practical clinical cases. Moreover, it would be convenient to undertake a more detailed study based on calculation software using the finite element method to obtain a more precise notion of the stresses to which the prosthesis is subjected. A topological optimisation of the geometry using suitable software could also be useful.

The current proposal would be equivalent to a TRL2 prototype according to the well-known TRL scale applied to technology, whose value (from 1–9) gives an indication of the level of maturity of the product, which implies that it is the formulation of a concept that has not yet been validated in the laboratory. Therefore, our proposal considers the mandibular prosthesis concept like an embryo, without practical validity, although its correct evolution and development could mean important advances in the field of intervention for mandibular fractures, not only in cats, but also in other species, by avoiding complications caused by damaging the integrity of the tooth roots and neurovascular supply when placing the screws to fix the jaw prosthesis.

This prosthesis model aims to visually reveal the proposals and ideas expressed throughout this review, without delving into the details about main dimensions, materials, manufacturing process and other aspects that may affect its implementation. Prostheses of these characteristics should be designed specifically for each specimen, adapted in a 3D model of the cat patient’s jaw. In this way, not only will the size be adjusted, but the surfaces will also adapt much better, and thus a better grip will be ensured with all the advantages that this entails for fracture healing.

## 5. Future Trends

On one hand, the use of virtual surgical planning (VSP) and CAD/CAM (Computer-Aided Design/Computer-Aided Manufacturing) technology has added a new dimension to surgical planning, especially in the areas of craniomaxillofacial trauma, orthognatic surgery and reconstructive maxillofacial surgery [32]. This technology allows increased accuracy of reconstruction, decreased operative time, decreased flap ischaemic times, ease of use, improved predictability of outcomes, improved patient satisfaction and decreased complications [30]. In addition, with the aid of the surgical cutting guide, the positioning and fixation of the prosthesis would be accurate and almost flawless, thereby greatly reducing the operating time [32].

On the other hand, a comparison of three cone-beam computed tomography (CBCT) methods with dental radiography was analysed by Heney et al. [33]. They found that CBCT methods were better suited than dental radiography to the identification of anatomical structures in the full mouth of cats [15,21]. Cone-beam CT may prove to be the next major advancement in veterinary dentoalveolar and maxillofacial imaging because of its ability to provide 3D imaging at a lower cost than conventional CT, and with a lower radiation risk. The use of rapid scan technology, which allows faster image acquisition than conventional CT, and the ability to post-process the volumetric data into various two-dimensional (2D) and 3D reconstructions, makes CBCT an attractive imaging modality [33].

Liptak et al. [34] were one of the first to report a partial reconstruction of the mandibular body with a customised 3D-printed titanium prosthesis in a cat after removal of a mandibular osteosarcoma, as mandibulectomy in cats is associated with a high complication rate [35], including short-term (<4 weeks) and long-term (>4 weeks) adverse effects such as dysphagia or inappetence, mandibular drift or malocclusion. To avoid anorexia after surgery due to pain, supplemental tube feeding is recommended following mandibulectomy in cats. However, almost one in eight of the cats in the study never returned to voluntary eating after mandibulectomy [35]. Liptak et al. [34] pointed out that, in cats and dogs, dental roots and neurovascular structures comprise most of the bone volume in the mandibles and maxilla, and avoiding these structures is important during mandibular and maxillary repair. This is a further challenge in the rostral portion of the mandible because the canine dental root fills most of the mandible. The rationale for avoiding the dental roots is the high likelihood of tooth death and consequent periapical periodontitis, resulting in infection and potential implant failure. In general, this case is a good example of the future trends in cat mandibular osteosynthesis. However, we consider that surgeons would have been more concerned about the following issues: (1) neurovascular bundle and (2) the screw size used in this cat patient. Regarding the neurovascular supply, the authors did not report the fate of the neurovascular bundle in the mandibular canal nor what they did with these structures before/during/after cutting the mandibular body [34]. Also, in this case report, a bone fragment was kept in the rostral part of the mandible to fix the prosthesis (after a mandibulectomy of 40 mm in the mandibular body). It could be expected that lacking ipsilateral blood supply and innervation, this fragment would not survive. On the contrary, it showed no evidence of failure 14 months postoperatively. It is reasonable to think that this bone fragment might have achieved collateral blood supply from the mandibular symphysis vessels due to tissue hypoxia that may have promoted angiogenesis. However, in our opinion, surgeons must be more attentive to the importance of avoiding damage to the neurovascular bundle because new blood vessels may grow from pre-existing ones (angiogenesis), but the sensitive (and motor) neural components do not recuperate and could cause additional neural damage. This is important to the well-being of the cat. Regarding point (2), the 2.0 mm-diameter screws used to fix the custom-made prosthesis seemed too large and long for the cat mandible, and after traversing both bone cortexes, their tips would become embedded into other structures, such as muscles or damage any intermandibular structure. Hence, using a screw size suitable to the cat patient is highly recommended.

On one hand, the mandibular prosthesis size should be adapted to the small size of feline patients. It seems that most pet prostheses are designed for bigger specimens (except those designed for a specific cat patient, printed in titanium), they must have no sharp edges, and finally, they should have plenty of holes to place the screws and for muscle or tissue attachment. However, the screws used are usually too long, far surpassing the contralateral cortical bone.

On the other hand, materials used in osteosynthesis must be more biologically acceptable because soft tissue is not going to attach to metal [36], so a deeper understanding is needed, and further research will be required to produce more biocompatible prostheses.

To close this section, we cannot agree more with Vaughan [36], who stated “custom-shaped implants will likely be part of the future, but the process needs to be refined from a biomechanical and biological perspective”.

## 6. Conclusions

In the body of the cat mandible, dental roots and the mandibular canal (with the vascular supply and the inferior alveolar nerve) occupy most of the volume. Therefore, in mandibular fractures (due to a variety of causes, such as periodontitis, tooth resorption, trauma, or secondary to tooth extraction), it makes it challenging to apply a plate with fixed screw positions without invading dental roots or neurovascular structures. Otherwise, it would be a very painful process, including the failure of fixation due to chronic infection and inflammation at the fixation site. In the face of all these difficulties, we proposed a suitable prosthesis design, produced by additive manufacturing, that would provide acceptably rigid biomechanical stabilisation and avoid damage to any of those structures when fixing a mandibular body fracture. The future depends on the improvement of diagnostic images and Computer-Aided Design/Computer-Aided Manufacturing (CAD/CAM) technology to manufacture custom-designed prostheses made of highly biocompatible material.

## Figures and Tables

**Figure 1 animals-11-00683-f001:**
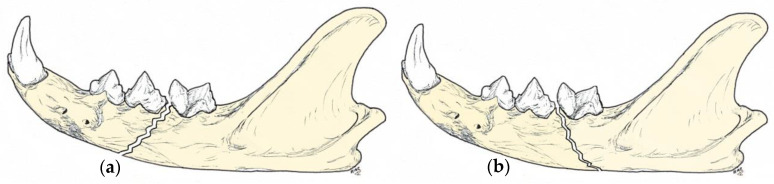
Drawings depicting the lateral view of the feline left mandible with a favourable fracture (**a**), as it compresses the fracture fragments, and an unfavourable fracture (**b**) in which the fracture segments are distracted.

**Figure 2 animals-11-00683-f002:**
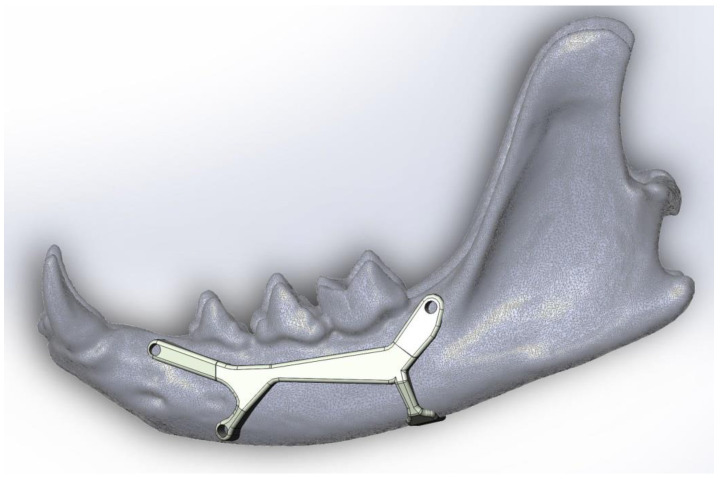
Lateral view of the mandibular prosthesis conceptual model in place, as the upper screws should be fixed where there are no dental roots. This model would be useful to repair body fractures between the third premolar and the first molar.

**Figure 3 animals-11-00683-f003:**
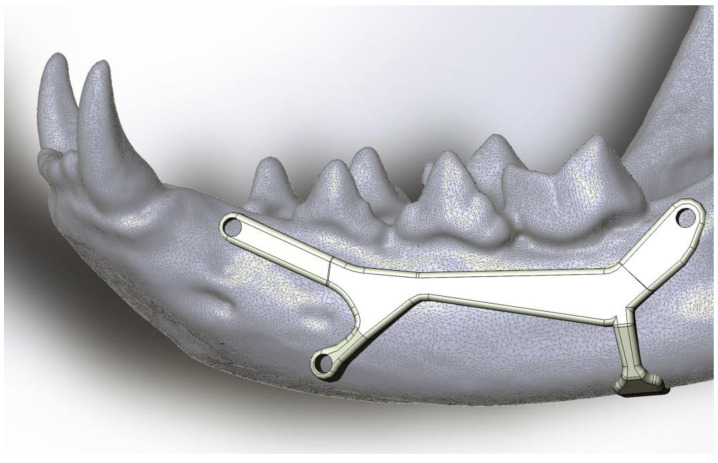
Magnification of Figure 2. The asymmetric shape of the anterior horizontal “Y” avoids damaging the nerves that come out through the main and mental foramina.

**Figure 4 animals-11-00683-f004:**
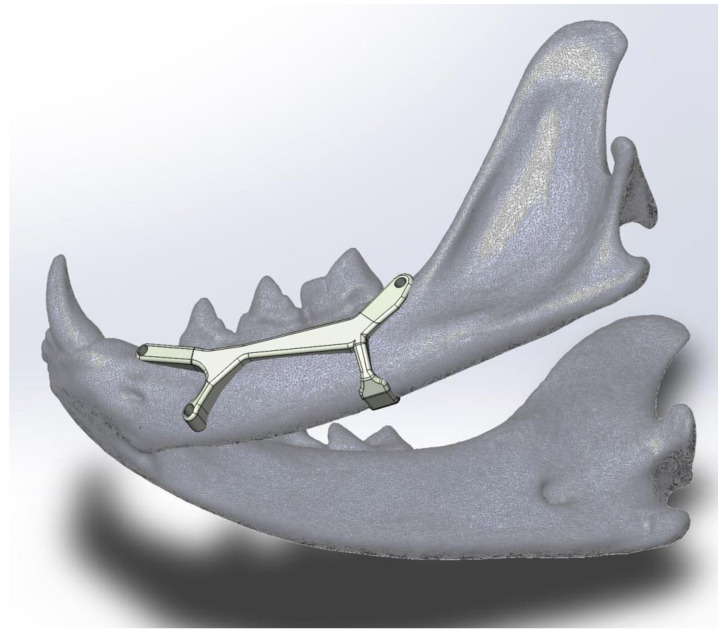
A ventrolateral view of the left jaw displaying the prosthesis conceptual model to show the fourth fixing point with no screw holes. This part consists of a flat hook-like device that surrounds and embraces the mandibular ventral margin to avoid damaging the neurovascular supply when drilling the mandibular canal. As this prosthesis is custom-designed, the flat hook size (thickness, width and length) will be variable, depending on the patient.

**Figure 5 animals-11-00683-f005:**
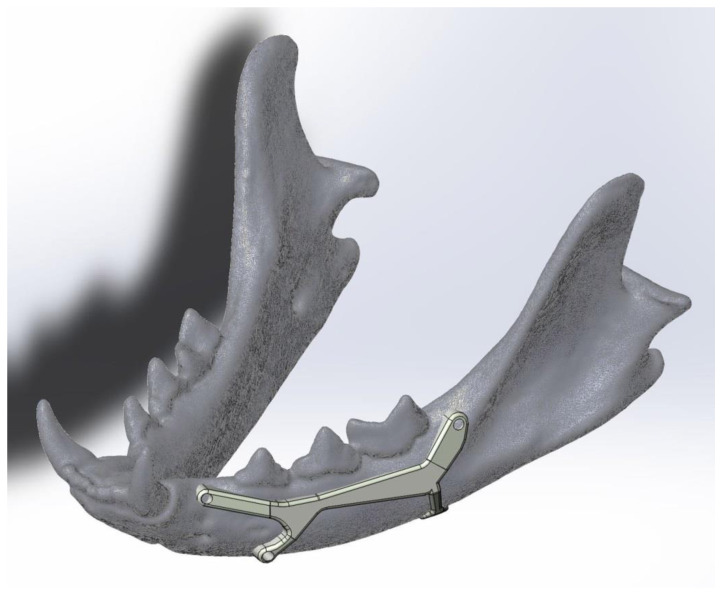
A frontolateral view of the left mandible with the proposed prosthesis. Note the prosthesis thickness is variable depending on the mandibular area in order to fully adapt to its contour. Prosthesis thickness does not exceed 1.2 mm at any point.

**Table 1 animals-11-00683-t001:** Mandibular fracture types. Their incidence, possible treatment methods and some recommendations are compiled.

	**Incidence (%)**	**Cause**	**Treatment Methods**	**Pay Special Attention to…**
Mandibular fractures in general	11–23 [4] 14,5 [10]	→ Road traffic accidents, fighting injuries, falls from heights, human abuse [4,5,6,7]. → Secondary to neoplasia, metabolic disease, and dental treatment [8].	→ A non-invasive treatment should be considered first [13]. → In multiple fractures, repair the mandible from caudal to rostral [17].	→ The naturally contaminated environment of the oral cavity. → Not damaging teeth and tooth roots. → Keep the integrity of the neurovascular supply in the mandibular canal. → Multiple or open fractures cause more complications [10]. → Infection and persistent periodontal disease can lead to osteomyelitis and non-union of fracture fragments [15].
**Mandibular Fracture Types**	**Incidence (%)** [10]	**Treatment Methods**	**Clinical Union (Weeks)** [10]	**Pay Special Attention to…** **Recommendations**
Symphyseal	73.3	→ Cerclage wire [4,15,26].	6 (3–12)	→ Be sure that the incisor teeth remain in alignment; otherwise, step defects can be generated [15].
Parasymphyseal		→ Osseous circumferential wiring [15].		→ Fracture is often non-displaced [19]. → Possibility of iatrogenic fracture after canine extraction when pre-existing periodontal disease, insufficient preparation prior to extraction or use of excessive force, or a combination [12,15,19]. → Do two radiographies: before and after dental extraction [20]. → Inform the client in advance that iatrogenic fracture is a possibility after canine extraction [20].
Body	16.0	→ Simple fractures (fracture line is perpendicular to the long axis of the mandible): internal fixation with interfragmentary wires [26]. → Oblique fractures: ◆ Favourable (mastication muscles compress the bone fragments, relatively stable): Conservative methods are enough to get stabilization of the tension surface [15,21]. ◆ Unfavourable (mastication muscles distract bone fragments): (A) Two intraosseus wires (one dorsally and the other ventrally) between the two fragments [15]; (B) Triangular method (two wires caudally to the fracture in a 90° angle to reach the same rostral hole [15]; (C) Internal fixation with conventional bone miniplates and screws [8,16]. → Open or comminuted fractures: external fixation with pins and mandibular bumper bar [26].	10 (8–16)	→ Keep the integrity of the neurovascular supply in the mandibular canal. → Avoid damaging the roots and periodontal ligament of the teeth [4,15]. → In-line tooth removal is not advised unless the teeth involved are fractured [8,17]. → Do postoperative radiographs, especially when severe periodontitis or other debilitating mandibular bone pathologies [19]. → The wiring must be placed so that its acts as a tension band to create interfragmentary compression [8]. → The most common complications of surgical repair are malocclusion and osteomyelitis [8]. → Ventral approach facilitates exposure, bone fragment reduction and stabilization [17]. → Use bone plates and screws in accordance with the cat mandibular size.
Ramus:				
Condylar process	6.7	→ Simple fractures heal by themselves as a functional and painless nonunion [3].	6 (4–8)	→ Comminuted fractures could generate TMJ ankylosis in young cats [3].
Coronoid process	4.0	→ A non-invasive treatment should be considered first [13].	6	

## Data Availability

Data sharing is not applicable to this article as all data associated is available in the text.

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
