# Peer review of "The Cat Mandible (II): Manipulation of the Jaw, with a New Prosthesis Proposal, to Avoid Iatrogenic Complications"

_animals, 2021, doi:10.3390/ani11030683_

Round 1

Reviewer 1 Report

In general this review manuscript entitled "The cat mandible (II): manipulation of the jaw, with a new prosthesis proposal, to avoid iatrogenic complications" is well written. 

My minor comments:

  • In section: "4. Prosthesis proposal to fix a simple fracture of the mandibular body" - please add more details about the preparing of all the prosthesis conceptual models (Figure 2, Figure 3, Figure 4, Figure 5)
  • In subsection: "4.1. Calculations"- please add the references how to perform the calculation (it is not clear the source of these formulas)

Reviewer 2 Report

Dear Authors,

the paper titled: The cat mandible (II): manipulation of the jaw, with a new prosthesis proposal to avoid iatrogenic complications reviewed the literature regarding the the use of fixing methods and different repair techniques in cats affected by mandibular luxation/fracture.

The paper is well written and does not require revisions but some contents needs to be addressed 

- please provide more details regarding the imaging technique for diagnosis as in you paper is missing.

- ln 29 - change "chirurgic" with surgical

- ln 250-263: rephrase this paragraph using more appropriate terms. As researchers, you are in title to disagree the outcome of an article but you can't provide negative comments (above all 259-261).

- All cap.4 needs to be deleted. I agree your idea looks very interesting but you are not providing any data for evaluation. For the same reason please rephrase  title, summary, abstract and document. 

Thank you

Reviewer 3 Report

Thank you for submitting. My comments were as follows.

1. Since “Prosthesis proposal to fix a simple fracture of the mandibular body” (Line 374-455) is unpublished content, I think this paper is not a review, but a commentary or article.

2. It is easier for the reader to understand the following contents in tables. 1) Fracture type and its incidence, treatment methods for each, 2) treatment methods and side effects of each fracture, 3) anatomical position and function of structures such as nerves and blood vessels that require attention in mandibular surgery.

Round 2

Reviewer 2 Report

Dear Authors,

thank you very much for your revisions.

You article is very interesting but as previously I am kindly asking to delete chapter 4.

As you say:

The proposed prosthesis (Figures 2–5) is a conceptual design; therefore, it would require further adjustment for specific clinical cases. Unfortunately, we have not yet tested it, but, in theory, we think it could achieve good results as it has three fixation points with small screws and the fourth is like a folded tab for fastening the ventral edge, so avoiding drilling the mandibular canal”.

Again, I think your idea is very interesting and I strongly hope you may propose further studies in which at least ex vivo, ideally in vivo tests may be achieved.

As you properly described mandibular fixation in cats is a very challenging procedure. Theoretically your prosthesis may be an alternative solution and I guess so, but we are not able to exclude any possible complications so far.

From my point of view the lack of this information is critical and needs to be proven.

Thanks again for your work

Reviewer 3 Report

Thank you for your reply. My comments were as follows.

Line 126-388 (chapter 3); I was expecting that the text would be summarized by creating a Table. The content of each reference is explained paragraph by paragraph. It will be easier to read if you explain according to each item such as fracture site, type, repair method, evaluation method, and complications, etc.

Table 1; The explanations written in “Treatment methods” of “Parasymphyseal fracture”, “Ramus fracture” and “Condylar rocess fracture” are not treatment methods.
